# Research and Design of a Medial-Support Exoskeleton Chair

**DOI:** 10.3390/biomimetics10050330

**Published:** 2025-05-18

**Authors:** Wenzhou Lin, Yin Xiong, Chunqiang Zhang, Xupeng Wang, Bing Han

**Affiliations:** 1School of Art and Design, Xi’an University of Technology, Xi’an 710054, China; lin2004@xaut.edu.cn (W.L.); 2220621001@stu.xaut.edu.cn (Y.X.); cqsay@xaut.edu.cn (C.Z.); 2School of Mechanical and Electrical Engineering, Northwestern Polytechnical University, Xi’an 710021, China

**Keywords:** medial support, exoskeleton chair, gait analysis, mechanical model

## Abstract

To address lower limb fatigue in workers engaged in prolonged standing, this study proposes a structural design for a medial-support passive exoskeleton seat. The design incorporates support rods positioned along the medial aspect of the user’s lower limbs and features an adaptive telescopic rod system, enhancing sitting stability and reducing collision risks in workplace environments. Human motion capture technology was used to collect kinematic data of the lower limbs, and a mathematical model of center-of-gravity variation was developed to calculate and optimize the exoskeleton’s structural parameters. Static analysis was performed using ANSYS software (2025 R1) to evaluate the structural integrity of the design. The effectiveness of the exoskeleton seat was validated through surface electromyography (sEMG) experiments, with results showing that the exoskeleton significantly reduces lower limb muscle load by 49.2% to 72.9%. Additionally, force plate experiments demonstrated that the exoskeleton seat improves stability, with a 39.2% reduction in the average displacement of the center of pressure (CoP), confirming its superior postural alignment and balance. The design was also compared with existing exoskeleton chairs, showing comparable or better performance in terms of muscle load reduction, stability, and overall effectiveness.

## 1. Introduction

Workers in occupations requiring prolonged standing (e.g., assembly line operators, healthcare staff) exhibit reduced operational efficiency due to lumbar and lower limb muscle fatigue [1]. Prolonged orthostatic stress without sufficient rest intervals elevates the risk of occupational disorders including varicose veins, osteoarthritis, and chronic musculoskeletal injuries, ultimately shortening career longevity [2]. A representative study in a Guangdong furniture manufacturing facility (China) reported a 32.12% prevalence of work-related musculoskeletal disorders [3]. These findings necessitate the integration of ergonomic equipment in industrial workflows to mitigate fatigue-induced productivity losses.

Wearable exoskeletons chairs employ kinematic support mechanisms to offload lower limb muscular loads while preserving mobility. Unlike static seating solutions, these devices integrate biomimetic articulations enabling ambulatory transitions between standing and seated positions via posture-adaptive control systems [4]. For operators in constrained workspaces requiring frequent vertical adjustments, such dynamic postural stabilization reduces cumulative fatigue and mitigates venous stasis risks. The system’s portability further supports task continuity across multiple worksites without restricting operational workflows. By synergistically addressing ergonomic stressors and musculoskeletal injury prevention, this technology demonstrates significant potential for enhancing both occupational health and industrial productivity.

Exoskeleton chairs are broadly classified into active and passive variants based on actuation methodologies [5]. Active systems employ powered actuators (e.g., electromechanical drives, hydraulic/pneumatic actuators) to generate assistive torque, while passive counterparts utilize elastic elements (springs, compliant mechanisms) to store/release mechanical energy during sit–stand transitions. The latter’s superior mass efficiency, cost- effectiveness, and mechanical adaptability make them particularly viable for dynamic industrial settings with spatial constraints. Commercial implementations like NOONEE’s Chairless Chair [6] demonstrate robust posterior load transfer capabilities through optimized spring kinematics, while Hyundai’s CEX exoskeleton [7] validated 30.59~84.08% muscular activity reduction via surface electromyography in automotive assembly tasks. Similarly, the legX exoskeleton [8] was shown to reduce rectus femoris activity by up to 56% in panel work tasks through its spring-locking hybrid mechanism.

Academic innovations have further advanced this domain [9]: Feng et al. [10] proposed a bioinspired dual-mode exoskeleton enabling context-aware switching between upper/lower limb assistance through bistable actuators. Zlatko L and Marc D [11] designed a passive seat mechanism capable of assisting walking by applying an upward support force to the user’s pelvis. Han’s team [12] implemented a passive chair exoskeleton with extended support surfaces via redundant linkage configurations, achieving 27% improved weight distribution. Zhang’s medical exoskeleton prototype [13] incorporated reconfigurable joint assemblies permitting multi-postural stabilization across 0–120° hip/knee flexion ranges.

Current wearable exoskeleton chairs face two critical limitations: the limited ground support area requires users to continuously activate stabilizing muscles during seated phases, which paradoxically increases physical fatigue. Moreover, many exoskeleton chair designs for workers fail to account for the complexity of real-world workspaces [14], resulting in increased collision risks when worn in confined environments, thereby compromising both operational safety and efficiency.

To address these issues, this study presents a medial-support exoskeleton chair with auxiliary support rods. This design enhances postural stability while seated and optimizes the arrangement of exoskeleton components to improve adaptability in complex work environments.

## 2. Human Lower Limb Motion Data Collection

During exoskeleton-assisted sit-to-stand transitions and ambulation, the lower limb’s polycentric kinematic chain—comprising hip, knee, and ankle joints—exhibits coordinated triplanar movements. Although all three joints demonstrate triaxial mobility, sagittal plane kinematics dominate functional patterns, particularly in the knee joint, which bears primary weight-transfer responsibilities [15]. This articulatory hierarchy necessitates precise biomimetic alignment in exoskeletal joint design.

A NOKOV^®^ infrared optical motion capture system (v3.0) was employed and synchronized at 200 Hz to quantify physiological joint excursions. This high-fidelity motion tracking modality enabled millimeter-level resolution in recording lower limb articulation angles during transitional movements and steady-state gait cycles. Experimental protocols captured both dynamic sit-to-stand phase transitions and linear locomotion sequences, establishing baseline kinematic profiles for subsequent exoskeleton optimization.

During the experiment, reflective markers (mark points) were affixed to specific body locations of the subjects. As the subjects performed the designated movements within the experimental area, multiple cameras positioned around the scene captured the locations of the markers. The software then calculated key motion parameters such as position, velocity, and acceleration, thereby obtaining the motion data of the human body. The experimental setup is shown in Figure 1.

Subjects followed the instructions to perform either normal walking or squatting movements at specific angles, while the joint angle variations of the major lower limb joints (hip, knee, and ankle) were recorded. The experimental results are as follows:

The recorded data of hip, knee, and ankle joint angle variations during squatting movements are shown in Figure 2. The results indicate that during sit-to-stand motion, the hip joint angle varies from approximately 75° to 180°, the knee joint angle ranges from 90° to 170°, and the ankle joint angle changes between 75° and 100°.

According to the joint angle data of normal walking, as shown in Figure 3, the results indicate that during small-scale normal walking, the hip joint angle ranges from 160° to 180°, the knee joint angle varies from 130° to 170°, and the ankle joint angle fluctuates between 85° and 110°.

The collected data provide a valuable reference for determining the dimensions and design of the exoskeleton chair.

## 3. Structural Principles and Parameter Determination of the Medial-Support Exoskeleton Chair

### 3.1. Structural Principles of the Medial-Support Exoskeleton

Contemporary exoskeleton chair designs predominantly employ lateral or posterior support configurations. While lateral alignment facilitates sagittal plane joint congruency during ambulation, and posterior designs exploit seated center of gravity shifts for load transfer, both approaches exhibit elevated collision susceptibility in spatially constrained industrial environments. To mitigate this limitation, this study proposed a medial-support architecture that positions primary load-bearing structures along the lower limbs’ medial aspect. This configuration preserves postural stability while enhancing workspace negotiability through reduced spatial occupancy.

The medial-support paradigm demonstrates biomechanical advantages over conventional layouts. By aligning force transmission pathways with the natural ground reaction force vector (directed toward the body’s center of mass during standing), this design optimizes structural efficiency and reduces component stress concentrations [16]. Furthermore, mass distribution along the sagittal midline minimizes rotational inertia, which can, to some extent, address the issue of insufficient coordination in exoskeleton chair systems [17].

Compared to active exoskeleton chairs, passive exoskeleton chairs are lighter and more user-friendly [18]. Therefore, the medial-support exoskeleton chair proposed in this study has a passive design, utilizing springs as energy storage and damping components.

To ensure ergonomic compatibility, the degrees of freedom (DOF) of the exoskeleton must match the corresponding human joints [19], and the movement angles should not exceed the physiological joint limits. The primary load-bearing rods are designed with adjustable lengths to accommodate users of different body sizes. To address motion interference, the following solutions are implemented:

The exoskeleton chair’s rotation axes are aligned with the human joint axes in the sagittal plane to minimize motion interference.

The medial-support structure positions the device’s main body between the legs, effectively reducing collision risks in confined workspaces while enhancing turning stability by concentrating mass distribution. And the total weight of the exoskeleton is controlled within 5 kg to minimize the additional metabolic burden on the lower limbs.

Many existing exoskeleton chairs enhance stability by incorporating additional support rods. While this effectively expands the support range, it also increases the risk of collisions with the surrounding environment. To address this issue, an innovative variable support mechanism is introduced, as shown in Figure 4. This mechanism consists of three sliding components, multiple springs, and an L-shaped support rod that functions as an automatic support system. When standing or walking, the support rod remains retracted within the lower leg structure to prevent external collisions. Upon entering a seated posture, a crank-slider mechanism lowers the sliding component, triggering the automatic deployment of the support rod, thereby expanding the exoskeleton’s support range.

The mechanism uses the knee joint angle β as the control parameter. When *β* > 130° (based on gait analysis data), the sliding motion does not activate the support rod. However, when *β* ≤ 130°, the sliding mechanism triggers the support rod’s deployment.

### 3.2. Key Structural Parameters of the Medial-Support Exoskeleton

The support range is verified using a simplified human body model (Figure 5), where the torso–lower limb system is simplified as a rigid linkage system. The relationship between the center of gravity projection and the support surface is analyzed: when the projection falls within the support surface, the body remains balanced; otherwise, instability occurs. For computational simplicity, the model is analyzed within a 2D sagittal plane.

The established sagittal-plane sit-to-stand model is shown in Figure 5, where a Cartesian coordinate system is defined with its origin at the sagittal-plane rotation center of the ankle joint. Point A represents the sagittal-plane rotation center of the hip joint, with coordinates $\left(x_{A}\right.,\left.y_{A}\right)$; Point B represents the sagittal-plane rotation center of the knee joint, with coordinates $\left(x_{B}\right.,\left.y_{B}\right)$; and Point C represents the sagittal-plane rotation center of the ankle joint, with coordinates $\left(x_{C}\right.,\left.y_{C}\right)$. The human body is divided into four segments: torso segment, length $l_{1}$, mass $m_{1}$, center of mass $G_{1}$, coordinates $\left(x_{1}\right.,\left.y_{1}\right)$; thigh segment, length $l_{2}$, mass $m_{2}$, center of mass $G_{2}$, coordinates $\left(x_{2}\right.,\left.y_{2}\right)$; shank segment, length $l_{3}$, mass $m_{3}$, center of mass $G_{3}$, coordinates $\left(x_{3}\right.,\left.y_{3}\right)$; and foot segment, length $l_{4}$, mass $m_{4}$, center of mass $G_{4}$, coordinates $\left(x_{4}\right.,\left.y_{4}\right)$.

It is assumed that the arms remain close to the torso in a downward position. Additionally, ε1, ε2, ε3, and ε4 represent the proportional distances of the segmental centers of mass from their proximal ends.

The coordinates of Point C (ankle joint) are expressed as:(1)$$\left\{\begin{array}{c} x_{C}=0\\ y_{C}=0 \end{array}\right.$$

The coordinates of the foot center of mass $G_{4}$ are expressed as:(2)$$\left\{\begin{array}{c} x_{4}=-l_{4}\varepsilon_{4}\\ y_{4}=0 \end{array}\right.$$

Given the ankle joint angle *α* and shank segment length $l_{3}$, the coordinates of Point B (knee joint) are expressed as:(3)$$\left\{\begin{array}{c} x_{B}=-l_{3}\cos\;\alpha \\ y_{B}=l_{3}\sin\;\alpha \end{array}\right.$$

The coordinates of the shank center of mass $G_{3}$ are expressed as:(4)$$\left\{\begin{array}{l} x_{B}=-l_{3}\left(1-\varepsilon_{3}\right)\cos\alpha \\ y_{B}=l_{3}\left(1-\varepsilon_{3}\right)\sin\alpha \end{array}\right.$$

Given the knee joint angle *β* and thigh segment length $l_{2}$, the coordinates of Point A (hip joint) are expressed as:(5)$$\left\{\begin{array}{c} x_{A}=-l_{3}\cos\alpha +l_{2}\cos(\beta -\alpha )\\ y_{A}=l_{3}\sin\alpha +l_{2}\sin(\beta -\alpha ) \end{array}\right.$$

The coordinates of the thigh center of mass $G_{2}$ are expressed as:(6)$$\left\{\begin{array}{c} x_{A}=-l_{3}\cos\alpha +l_{2}\left(1-\varepsilon_{2}\right)\cos(\beta -\alpha )\\ y_{A}=l_{3}\sin\alpha +l_{2}\left(1-\varepsilon_{2}\right)\sin(\beta -\alpha ) \end{array}\right.$$

Given the hip joint angle *γ* and torso segment length $l_{1}$, the coordinates of the torso center of mass $G_{1}$ are expressed as:(7)$$\left\{\begin{array}{c} x_{1}=-l_{3}\cos\alpha +l_{2}\cos(\beta -\alpha )-l_{1}\left(1-\varepsilon_{1}\right)\cos(\gamma +\alpha -\beta )\\ y_{1}=l_{3}\sin\alpha +l_{2}\sin(\beta -\alpha )+l_{1}\left(1-\varepsilon_{1}\right)\sin(\gamma +\alpha -\beta ) \end{array}\right.$$

The overall center of mass position G is calculated as the mass-weighted vector sum of all segmental centers of mass:(8)$$\left\{\begin{array}{c} x_{G}=\overset{n}{\sum }\frac{m_{i}x_{i}}{M}\\ y_{G}=\overset{n}{\sum }\frac{m_{i}y_{i}}{M} \end{array}\right.$$
where *n* represents the number of body segments, *M* is the total body mass, and $\left(x_{i}\right.,\left.y_{i}\right)$ are the center coordinates of each segment.(9)$$x_{G}=\frac{-l_{4}\varepsilon_{4}m_{4}-l_{3}\left[m_{1}+m_{2}+\left(1-\varepsilon_{3}\right)m_{3}\right]\cos\alpha }{m_{1}+m_{2}+m_{3}+m_{4}}+\frac{l_{2}\left[m_{1}+\left(1-\varepsilon_{2}\right)m_{2}\right]\cos(\beta -\alpha )-l_{1}m_{1}\left(1-\varepsilon_{1}\right)\cos(\gamma +\alpha -\beta )}{m_{1}+m_{2}+m_{3}+m_{4}}\;+\frac{l_{2}\left[m_{1}+\left(1-\varepsilon_{2}\right)m_{2}\right]\cos(\beta -\alpha )-l_{1}m_{1}\left(1-\varepsilon_{1}\right)\cos(\gamma +\alpha -\beta )}{m_{1}+m_{2}+m_{3}+m_{4}}$$

The segmental mass ratios $\varepsilon_{i}$ and length coefficients $k_{i}=\frac{m_{i}}{m_{1}+m_{2}+m_{3}+m_{4}}$ can be obtained from *Research Methods in Biomechanics* [20]. The anthropometric dimensions of the lower limbs are referenced from the 2023 Chinese Adult Human Body Dimensions Standard (GB/T 10000-2023) [21], selecting the 90th percentile as design reference values, as shown in Table 1.

The reference values for human joint angle range of motion can be obtained from the previous human motion capture experiment. From Figure 3, it can be observed that during squatting, the ankle joint movement angle is approximately 30°, and the hip joint movement angle is about 110°. Therefore, when wearing the medial-support exoskeleton chair in a sitting posture, the ankle joint angle is set to $\alpha_{s}=70{}^{\circ}$, and the hip joint angle is set to $\gamma_{s}=\beta_{s}+10{}^{\circ}$. Assuming that the joint angle variation during the squatting process is linear, substituting the relevant data into Equation (9) yields the horizontal coordinate variation curve of the human center of gravity at knee joint angles of 120°, 110°, and 100°, as shown in Figure 6.

It is evident that the maximum value of the center of gravity coordinate appears when the wearer is fully seated. Therefore, by calculating the extreme values in Equation (9), the relationship between different knee joint support angles and the maximum value of the center of gravity coordinate can be obtained, as shown in Figure 7.

From Figure 7, it can be observed that as the knee joint support angle decreases, a longer support rod is needed to support the rearward-shifting center of gravity of the human body. However, the support rod length cannot exceed the length of the lower leg segment, so the knee joint support angle must be within a reasonable range. Figure 7 indicates that when the support angle exceeds 135°, the horizontal coordinate value of the center of gravity is less than 0, meaning that at this knee joint angle, the wearer can maintain balance without the need for exoskeleton support. Therefore, support angles beyond 135° are not considered. According to the human motion capture experiment, the minimum knee joint motion angle during normal walking is 130°, so the exoskeleton support angle should be less than this value. Thus, a support angle range of approximately 100° to 120° is deemed reasonable.

In summary, this study selects 110° as the primary support angle, with adjustable angles of 105° and 115°. The support rod provides a support range length of approximately 110–140 mm.

Figure 8 presents a schematic diagram of the exoskeleton structure in a seated posture, where $\theta_{1}=130{}^{\circ}$ and $\theta_{2}=110{}^{\circ}$, AB represents the length of the thigh rod, BC represents the length of the lower leg rod, DE represents the length of the connecting rod, denoted as $l_{c}$, and EF represents the length of the support rod, denoted as $l_{s}$. To ensure the support performance of the exoskeleton, the connection between the exoskeleton’s connecting rod and the thigh rod should be as close as possible to the center of gravity of the human thigh segment:(10)$$BD=\left(1-\varepsilon_{2}\right)BA$$

To determine the length of the support rod $l_{s}$, first obtain the movement distance of the connecting rod slider $S=EE^{\prime }$ when two angles change:(11)$$\left\{\begin{array}{c} 2DB\cdot BE\cos\theta_{2}=DB^{2}+BE^{2}-l_{c}^{2}\\ 2DB\cdot BE^{\prime }\cos\theta_{1}=DB^{2}+BE\prime^{2}-l_{c}^{2}\\ EE^{\prime }=BE-BE^{\prime } \end{array}\right.$$

Substituting this equation into MATLAB (R2024b) and inputting the data yields the relationship between S and $l_{c}$, as shown in Figure 9.

The support rod fully extends when the knee joint angle reaches the support angle and fully retracts when the limiting angle is not reached, given by:(12)$$EF+s=E^{\prime }C$$
where *s* represents the movement distance of the slider on the lower leg rod. This slider assists in extending and retracting the support rod and ensures full contact with the ground when fully seated. To maintain stability, the support point of the support rod and lower leg rod should be as close as possible to the center of gravity of the human lower leg segment when fully seated:(13)$$EC=\left(1-\varepsilon_{3}\right)BC$$

Substituting this condition into Equation (11) and solving for $l_{c}=461\mathrm{mm}$ and S=55mm, the results are used as the design dimensions. These values are then substituted into Equations (10)–(12), and solving further using Equation (11) yields a support rod length of 218 mm and *s* = 30 mm.

In conclusion, the calculated structural dimensions for each mechanism rod of the exoskeleton chair are as follows: thigh rod (length from knee joint to support point) 318 mm, lower leg rod 405 mm, connecting rod length 460 mm, support rod length 218 mm, lower leg rod slider movement distance 30 mm, and connecting rod slider movement distance 55 mm. The structural design and computer modeling are carried out based on these dimensional data.

## 4. Design and Strength Analysis of the Seat

### 4.1. Design of the Medial-Support Exoskeleton Chair

The wearable exoskeleton chair was designed using SolidWorks (2025), and its 3D model is shown in Figure 10. The core structure consists of the lower leg rod, thigh support rod, support rod, and multiple adjustment components. Figure 10a presents the exoskeleton in a standing posture, while Figure 10b illustrates the seated mode.

To enhance ergonomic adaptability, the exoskeleton chair included a length adjustment system. The support component at the end of the lower leg rod, which contacts the ground, had an adjustable mounting position to accommodate users with different leg lengths. The thigh rod incorporated a connector with multiple adjustable holes, allowing personalized control of the support angle when seated.

Additionally, to ensure a comfortable and secure fit to the lower limbs, multiple strap slots were integrated into the thigh and lower leg rods. These slots helped fix the straps, preventing displacement during movement. The upper thigh rod slot extended to a waist strap, preventing the device from slipping down during activities. The seat surface was contoured to fit the curve of the posterior thighs, improving sitting comfort.

The dynamic support mechanism featured a support rod positioned at the lower rear of the lower leg rod, using a three-slider linkage mechanism (Figure 11) to achieve automatic switching between two states: standing mode and walking mode. Figure 11a shows how the support rod remains folded against the lower leg to prevent collisions. The upper slider stays at the top of the sliding axis, with movement restricted to a knee joint angle range of β ≥ 130°, ensuring normal gait was unaffected. In sitting mode (Figure 11b), when the knee angle β < 130°, the upper slider presses the middle slider, triggering the lower slider to deploy the support rod. Once fully seated, the support rod touches the ground, forming an expanded support surface, improving sitting stability.

A compression spring was installed beneath the lower slider to absorb knee joint loads during sitting and release stored energy to assist standing. A dual-stage buffering system was incorporated: the upper spring (low elasticity modulus) and lower spring (high elasticity modulus) work together to ensure the support rod deploys first, then move downward to touch the ground, avoiding premature contact that could cause inconvenience.

### 4.2. Static Finite Element Analysis Verification

To verify the structural safety of the design, ANSYS finite element analysis (FEA) was performed on the computer model of the medial-support exoskeleton chair. The material selected was 7050-T7651 aluminum alloy steel. Considering that the hip joint normally bears approximately 60% of the body weight [22], the force applied to the thigh rod was set to *F* = 300 N.

The FEA results are shown in Figure 12, revealing a maximum static stress of 335.28 MPa and a minimum safety factor of 2.01, indicating structural safety within the material limits. These results confirm that the exoskeleton chair provides safe and reliable support for users.

## 5. Experimental Validation

Based on the established computer model, a physical prototype of the medial-support exoskeleton chair was fabricated using 3D printing technology. Additionally, an effectiveness verification experiment based on muscle electromyography (EMG) detection was designed. The physical prototype is shown in Figure 13, where Figure 13a presents the front view and Figure 13b presents the side view.

The surface electromyography (sEMG) detection system in the Noraxon Biomechanics Experimental System captures EMG intensity signals via sensors placed on the surface of the muscles. The muscle activity level can be evaluated based on the EMG signal intensity [23]. In the experiment, the test subject performed squatting movements under three conditions: without wearing the exoskeleton chair, wearing the exoskeleton chair with the rear support rod removed, and wearing the complete exoskeleton chair with support rods. By comparing the EMG intensity of lower limb muscles in these three scenarios, the effectiveness of the exoskeleton chair was assessed.

As shown in Figure 14, EMG sensors were placed on four highly engaged lower limb muscles during squatting: the rectus femoris, vastus lateralis, tibialis anterior, and lateral gastrocnemius.

Five healthy subjects (3 males and 2 females, mean age 24.8 ± 1.3 years, mean weight 63.0 ± 6.1 kg, mean height 172.6 ± 4.9 cm) participated in the experiment. The study was approved by the Ethics Committee of Xi’an University of Technology, and all participants provided written informed consent prior to the experiment. The experiment was divided into two stages: (1) surface electromyography (sEMG) measurements of lower limb muscles during short-term sitting, and (2) plantar pressure testing during prolonged sitting.

### 5.1. EMG Testing

Before formal testing, each subject performed maximum voluntary contractions (MVC) for each monitored muscle group following standardized protocols [24]. During the formal experiment, each subject performed five squatting movements under each of the three conditions: without the exoskeleton chair, with the exoskeleton chair but without the support rod, and with the complete exoskeleton chair. A 60 s rest period was provided between each movement to prevent muscle fatigue from affecting the experimental results.

The collected EMG signal data underwent filtering to eliminate noise interference. The signals were then rectified, and a 10 s segment of effective signal time was extracted. The root mean square (RMS) algorithm (100 ms window) was used to best display the shape of the activation pattern [25], then the data were normalized as a percentage of MVC. The processed values were finally averaged across trials for each condition. This normalization approach allows for meaningful comparison of muscle activation levels between different subjects [26]. The final experimental results are shown in Figure 15, which compares average muscle activity in four lower limb muscles under the three different experimental conditions: without the exoskeleton chair, with the exoskeleton chair but without the support rod, and with the complete exoskeleton chair.

### 5.2. Plantar Pressure Testing

In addition, to evaluate the stability of the proposed exoskeleton chair during prolonged sitting, a plantar pressure experiment was conducted while wearing the prototype. The plantar pressure can effectively reflect body stability during both sitting and standing movements [27]. In the experiment, subjects wore the exoskeleton chair and remained seated for 5 min under two conditions: with and without the extra support component. Each session included a 2 min break, and the experiment was repeated in three trials. A three-dimensional force plate was used to record the plantar pressure data throughout the trials. The force plate is equipped with four pressure sensors capable of measuring forces in three directions, with a sampling rate of 80 Hz. Figure 16 shows a schematic diagram of the 3D force plate used for plantar pressure testing, while Figure 17 illustrates the experimental procedure.

The collected data were processed using Equation (12) to calculate the coordinates of the center of pressure (CoP) at each time point. The stability of the sitting posture was assessed by analyzing the displacement of the CoP throughout the experiment.(14)COPx=a2Fz+ARFz−PRFz−ALFzPLFz+ARFz+PRFz+ALFzPLCOPy=b2Fz+ARFz−PRFz−ALFzPLFz+ARFz+PRFz+ALFzPL

The results are presented in Figure 18 and Figure 19, which display density maps of the CoP distribution under the two experimental conditions: with and without the extra support structure. The quantitative analysis revealed that the average displacement of the CoP decreased by 39.2% with the addition of the designed support system.

## 6. Assessment

The sEMG experimental data revealed that, when wearing the exoskeleton chair without the support rod, the muscle loads of the rectus femoris, vastus lateralis, tibialis anterior, and lateral gastrocnemius decreased by 65.2%, 23.3%, 63.1%, and 31.2%, respectively. In contrast, when wearing the complete exoskeleton chair with support rods, the muscle loads of these four muscles decreased further to 72.9%, 62.3%, 67.2%, and 49.2%, respectively.

It can be observed that the proposed medial-support exoskeleton seat significantly reduces lower limb muscle load during short-term squatting, with particularly notable reductions in the vastus lateralis oblique muscle and lateral gastrocnemius. Further comparative experiments revealed that the complete exoskeleton seat further reduced lower limb muscle load compared to the medial-support exoskeleton seat without the support rod system (rectus femoris by 7.7%, vastus lateralis oblique by 39.0%, tibialis anterior by 4.1%, and lateral gastrocnemius by 18.0%). This indicates that the support rod significantly alleviates lower limb muscle load during sitting by increasing the ground contact area.

The plantar pressure experiment results show that, without the support component, the CoP distribution is more scattered and uneven. Subjects tended to lean forward to maintain stability during prolonged sitting while the support system is unavailable, resulting in a forward-shifted center of mass. In contrast, when the full exoskeleton chair with the support system was worn, the CoP distribution was more concentrated and shifted posteriorly. Furthermore, the 39.2% reduction in CoP displacement also indicates that the extra support system design significantly enhances postural stability during prolonged sitting.

In Table 2, the key parameters and evaluation results of the proposed medial-support exoskeleton chair (including support method, main material, weight, maximum supported body weight, and muscle load reduction outcomes) are compared with five other different existing passive exoskeleton chairs.

The comparison results indicate that the proposed device demonstrates solid overall performance across several key parameters.

First, in terms of structural design, the device adopts a medial-support configuration, which is relatively uncommon compared to the lateral or posterior support approaches used in most existing devices. This design offers advantages such as a more compact structure and potentially greater comfort, reflecting a degree of structural innovation. In terms of materials, the device uses aluminum alloy as the primary structural component, resulting in a relatively low weight of 3.9 kg, placing it in the mid-to-lightweight range among comparable products.

Regarding load-bearing capacity, the device can support up to 100 kg, which is sufficient for the majority of adult users. Although this is not the highest among the compared devices, the load capacity is considered acceptable given the relatively low weight of the device, indicating a favorable balance between mass and support strength. In terms of muscle load reduction, evaluated through surface electromyography (EMG), the device achieves a reduction of 49.2% to 72.9%, which is close to the performance of most compared devices, suggesting a significant effect in reducing muscle fatigue.

It is worth noting that the proposed device is still at the prototype stage and not yet a mature commercial product. Nevertheless, its performance indicators have already shown a promising level of acceptability.

In conclusion, the exoskeleton chair developed in this study demonstrates good potential in terms of structural innovation, performance balance, and ergonomic effectiveness, providing a valuable technical reference for future optimization of exoskeleton seating systems.

## 7. Limitations and Future Work

Despite the promising performance demonstrated by the proposed exoskeleton chair, several limitations and challenges remain to be addressed in future research and development.

First, the current prototype was primarily designed to validate its structural functionality, with less emphasis on user comfort, aesthetic appearance, and ergonomic refinements. These aspects are crucial for practical adoption and long-term use, and thus require further iterative design and user-centered development.

Second, the experimental evaluation was conducted predominantly with younger adult participants, which may not fully represent the diverse range of potential users. Future studies should expand the participant pool to include individuals of various age groups and physical conditions. Additionally, further experiments are needed to assess long-duration usage, multi-task scenarios, and real-world applications, in order to better understand the device’s performance and usability in practical contexts.

Third, the prototype currently exhibits mechanical issues, such as suboptimal motion smoothness in certain joints and moving parts. This reflects an ongoing engineering challenge that must be addressed through improved mechanism design, tolerance control, and possibly the integration of higher-precision components.

In light of these limitations, future work will focus on enhancing both the mechanical performance and user-centered design, while conducting broader and more comprehensive experimental evaluations to support the device’s advancement toward real-world applicability.

## 8. Conclusions

This study proposed a medial-support exoskeleton chair design to address the stability limitations and structural collision issues of existing exoskeleton chairs. The key contributions of this study are as follows:

Structural Optimization: Various exoskeleton chair designs were analyzed and compared, leading to the selection of a medial-support structure to effectively reduce collision risks while wearing the exoskeleton.

Kinematic Analysis: Motion capture equipment was used to collect human movement data. Through mathematical modeling, changes in the lower limb center of gravity while wearing the exoskeleton were analyzed, and component dimensions were calculated to ensure freedom of movement and postural stability.

Structural Strength Validation: Finite element analysis (FEA) using ANSYS software was conducted to verify the structural strength and reliability of the exoskeleton chair.

Effectiveness Verification: A Noraxon biomechanics experiment was conducted using sEMG testing. The results demonstrated that the medial-support exoskeleton chair effectively reduces muscle load during squatting, and the additional support rod component provides enhanced stability, further decreasing lower limb muscle fatigue. The results demonstrated that the medial-support exoskeleton chair effectively reduces muscle load during squatting, with muscle load reduction ranging from 49.2% to 72.9%. Additionally, the extra support rod component provided enhanced stability, further reducing lower limb muscle fatigue. Furthermore, force plate experiments revealed that the CoP distribution became more concentrated and posteriorly shifted with the support system, leading to a 39.2% decrease in the average CoP displacement, indicating improved stability.

Additionally, this study also compared the proposed exoskeleton chair with several existing exoskeleton chairs. The comparison results showed that the medial-support exoskeleton chair performed well in terms of muscle load reduction, stability, and comfort, exhibiting comparable or better performance than other commercialized and prototype exoskeleton chairs.

This research provides valuable insights and optimization directions for future exoskeleton chair design improvements.

## Figures and Tables

**Figure 1 biomimetics-10-00330-f001:**
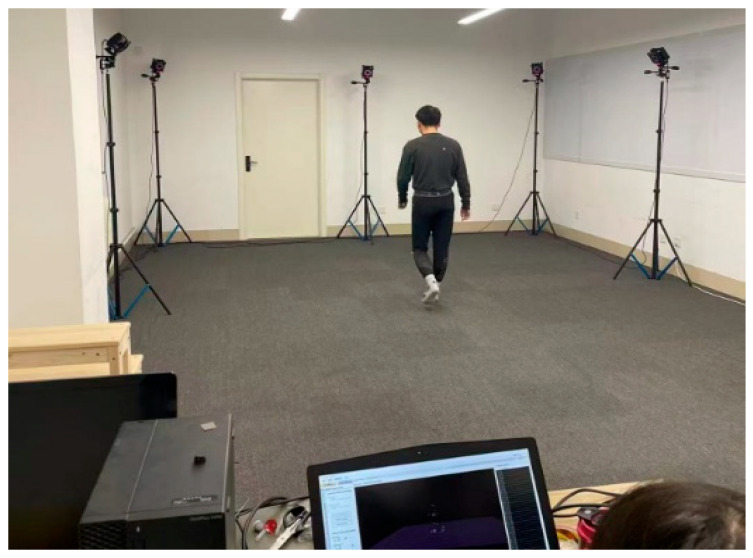
Human motion capture experiment.

**Figure 2 biomimetics-10-00330-f002:**
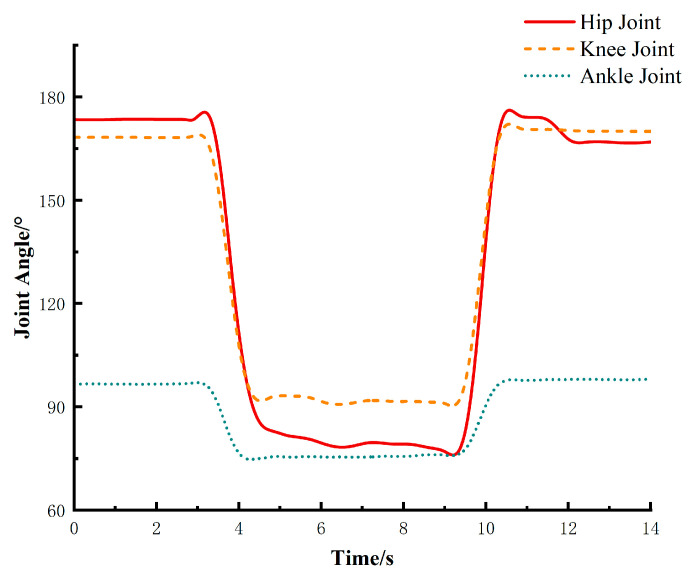
Changes in lower limb joint angles during sit-to-stand motion.

**Figure 3 biomimetics-10-00330-f003:**
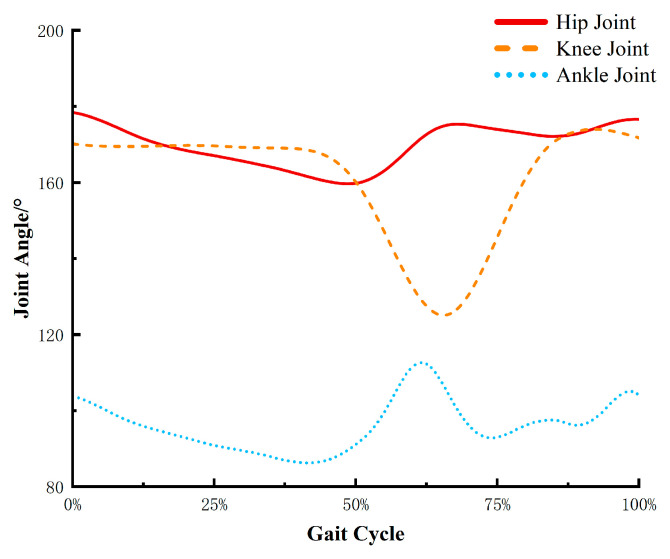
Changes in lower limb joint angles during normal human walking.

**Figure 4 biomimetics-10-00330-f004:**
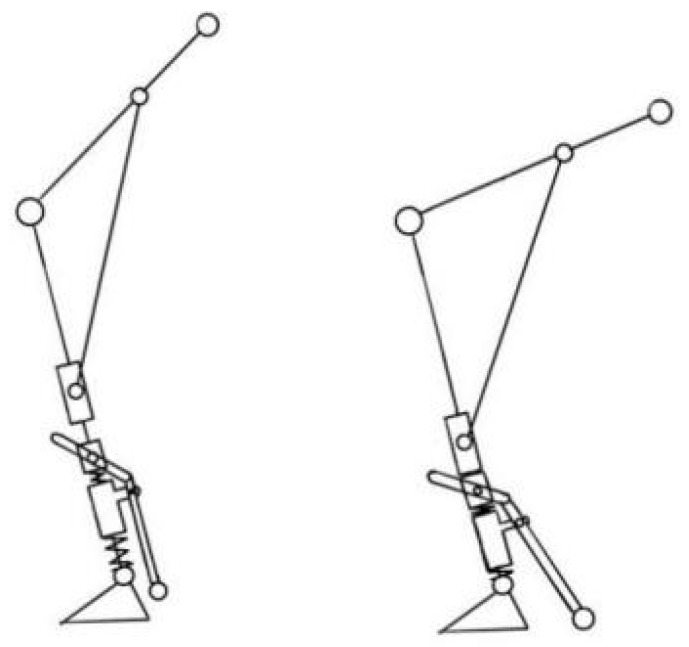
Schematic diagram of the exoskeleton support component structure.

**Figure 5 biomimetics-10-00330-f005:**
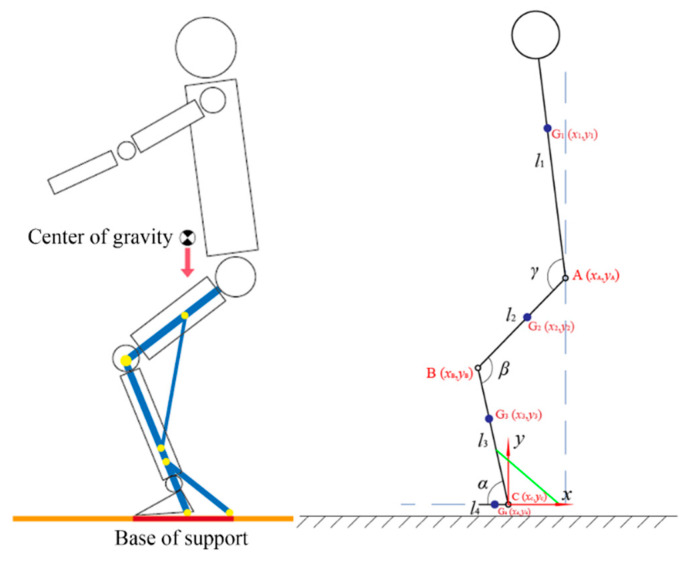
Sagittal-plane sitting and standing motion model.

**Figure 6 biomimetics-10-00330-f006:**
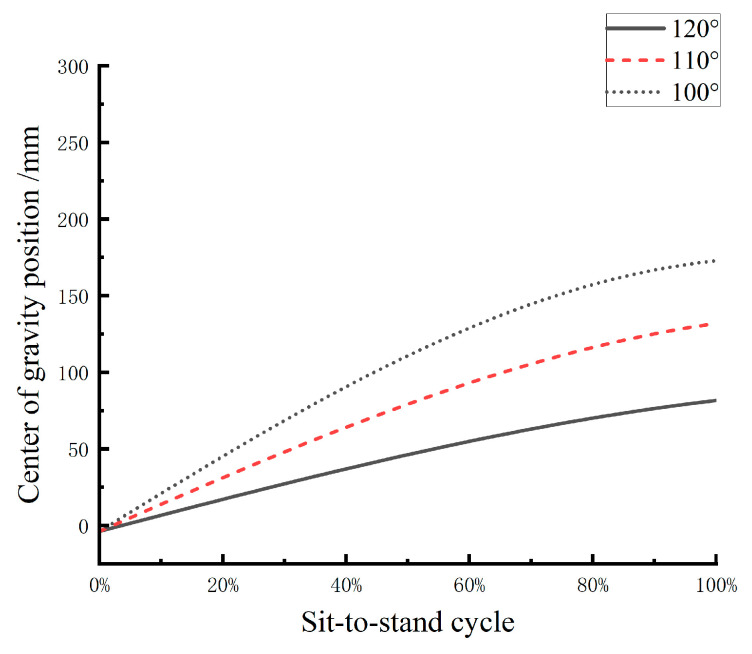
Variation of the human center of gravity in squatting posture at different knee angles.

**Figure 7 biomimetics-10-00330-f007:**
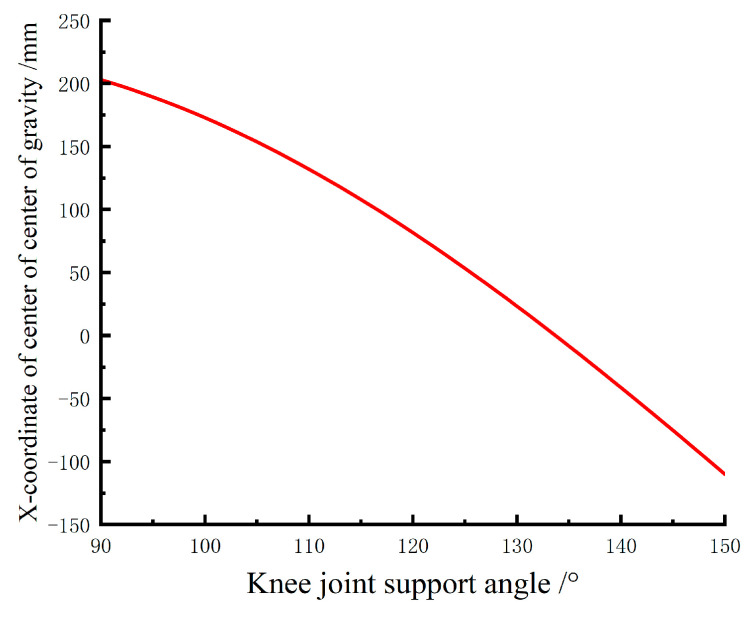
Variation of the maximum horizontal coordinate of the center of gravity at different knee angles.

**Figure 8 biomimetics-10-00330-f008:**
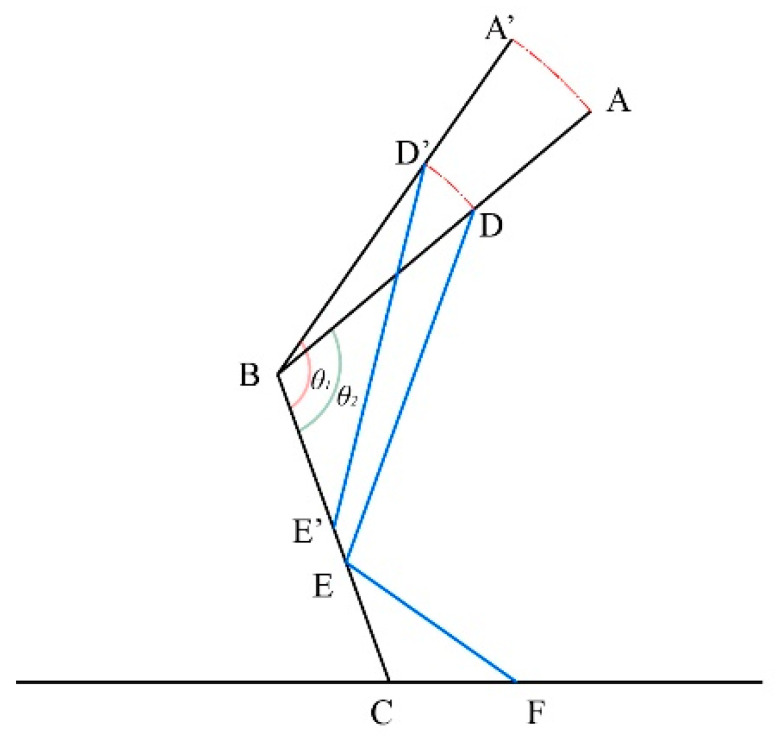
Schematic diagram of the exoskeleton seat structure in sitting posture.

**Figure 9 biomimetics-10-00330-f009:**
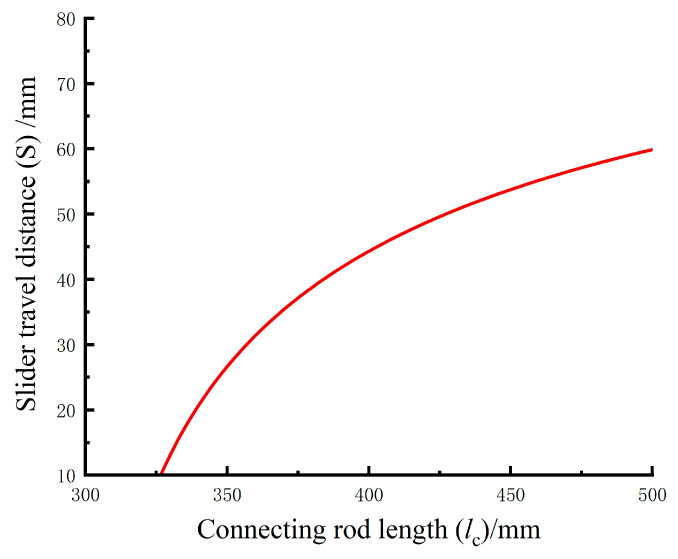
Relationship between the length of the connecting rod and the movement of the slider.

**Figure 10 biomimetics-10-00330-f010:**
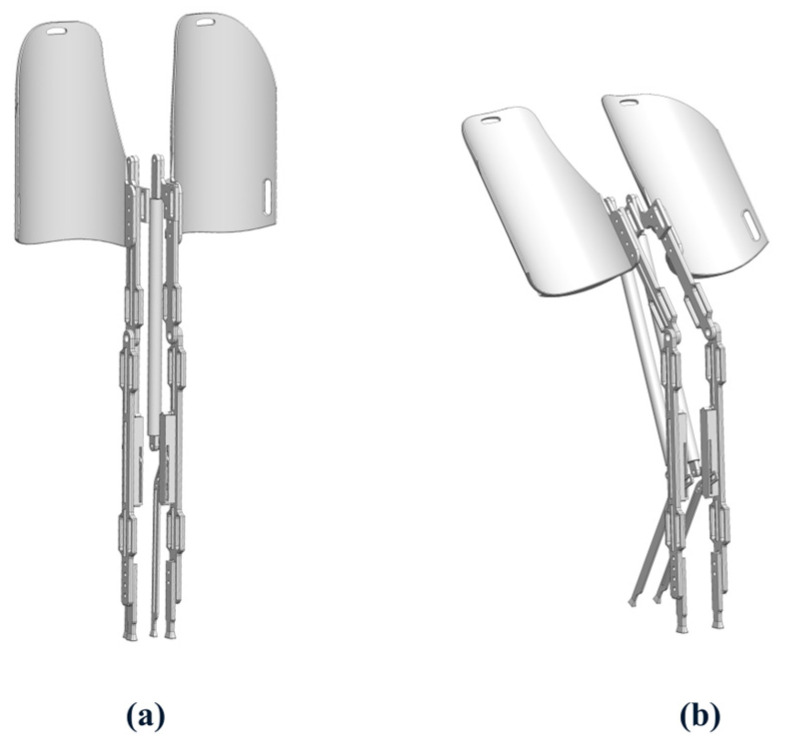
Model of the medial-support exoskeleton chair (**a**) Standing posture; (**b**) Sitting posture.

**Figure 11 biomimetics-10-00330-f011:**
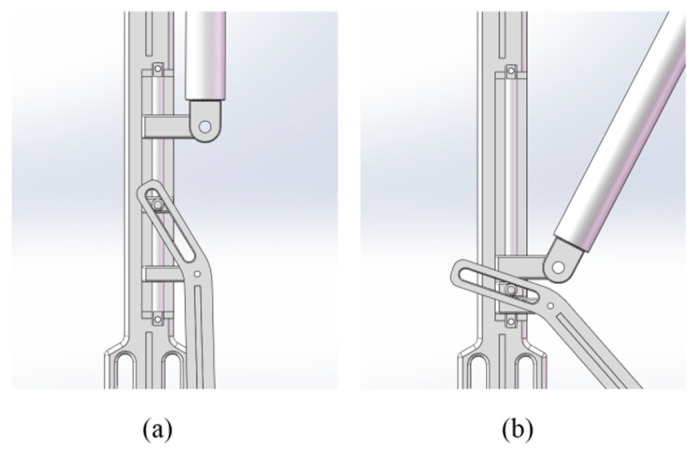
Schematic diagram of the motion model of the support rod: (**a**) standing; (**b**) sitting.

**Figure 12 biomimetics-10-00330-f012:**
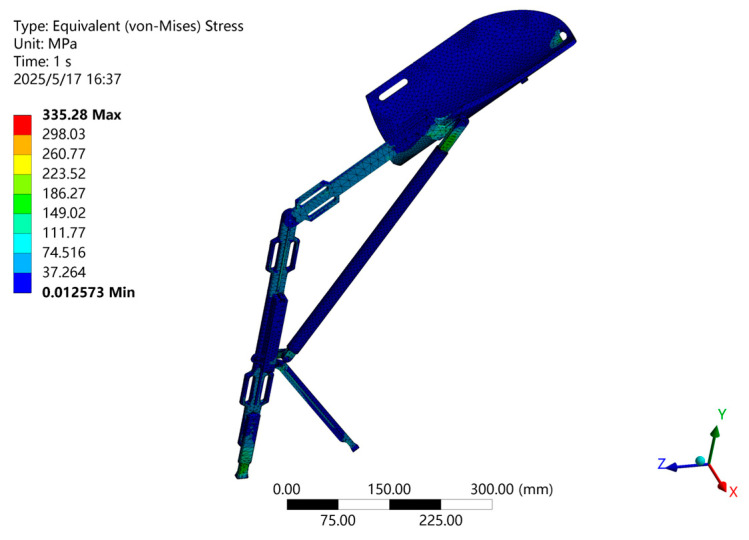
Static analysis results of the medial-support exoskeleton chair.

**Figure 13 biomimetics-10-00330-f013:**
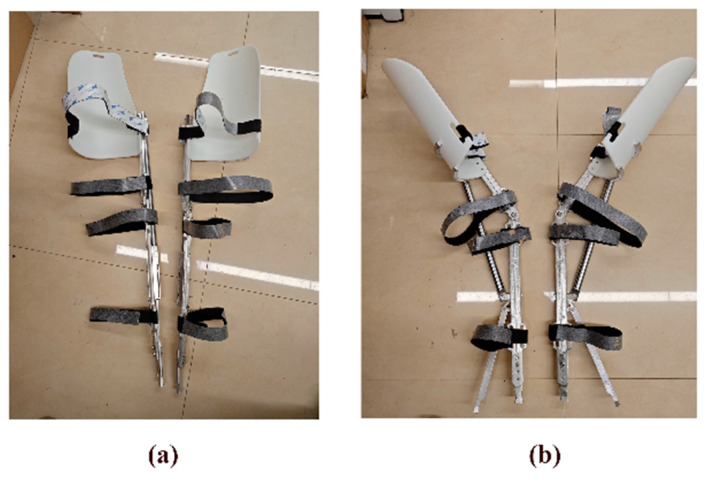
Exoskeleton seat prototype: (**a**) front; (**b**) side.

**Figure 14 biomimetics-10-00330-f014:**
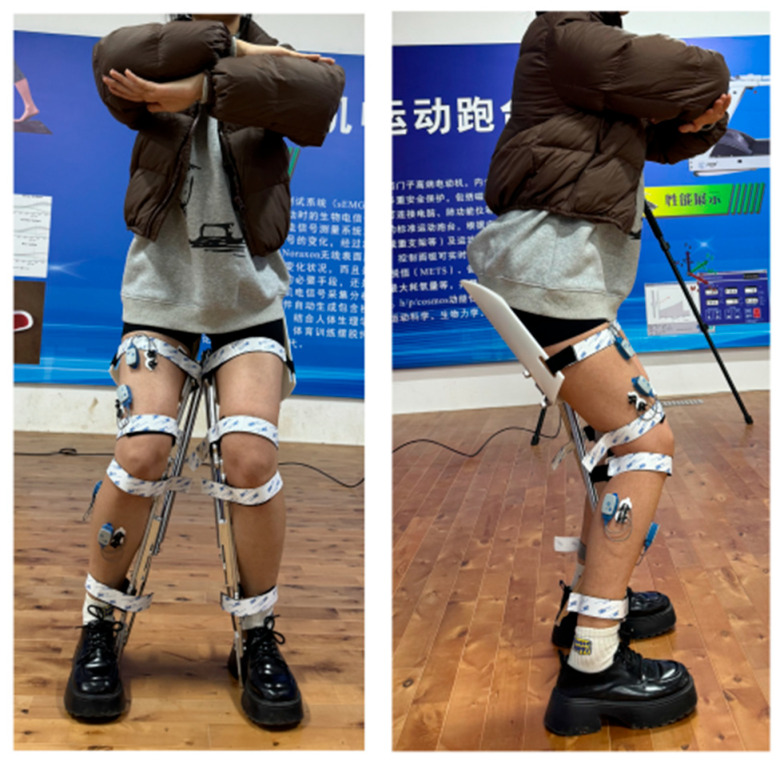
sEMG Testing.

**Figure 15 biomimetics-10-00330-f015:**
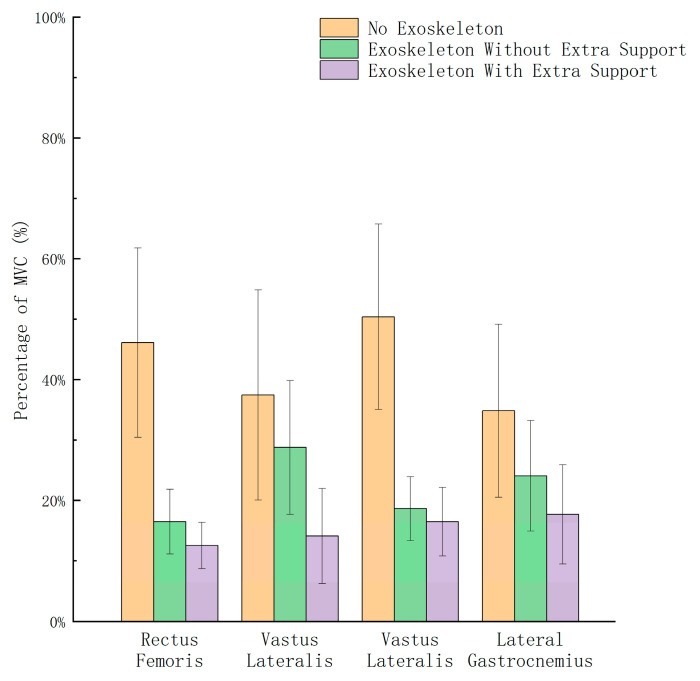
Experimental results of sEMG test.

**Figure 16 biomimetics-10-00330-f016:**
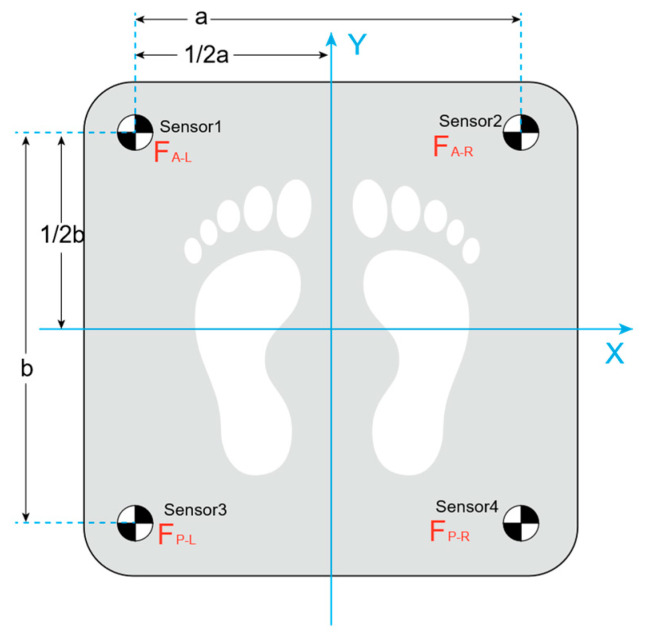
Dimensions and coordinates of three-dimensional force plate (a = b = 380 mm).

**Figure 17 biomimetics-10-00330-f017:**
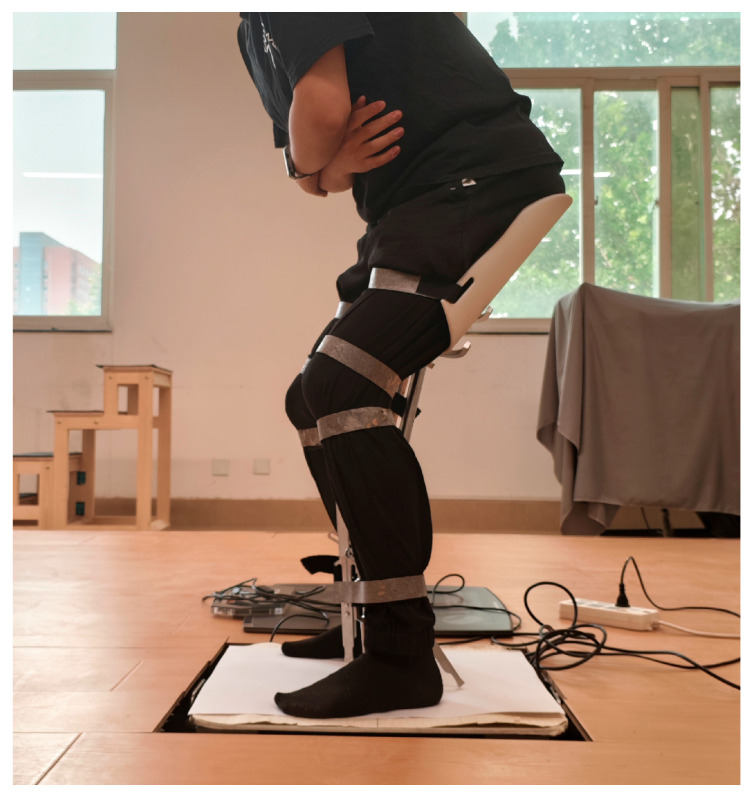
Plantar pressure testing.

**Figure 18 biomimetics-10-00330-f018:**
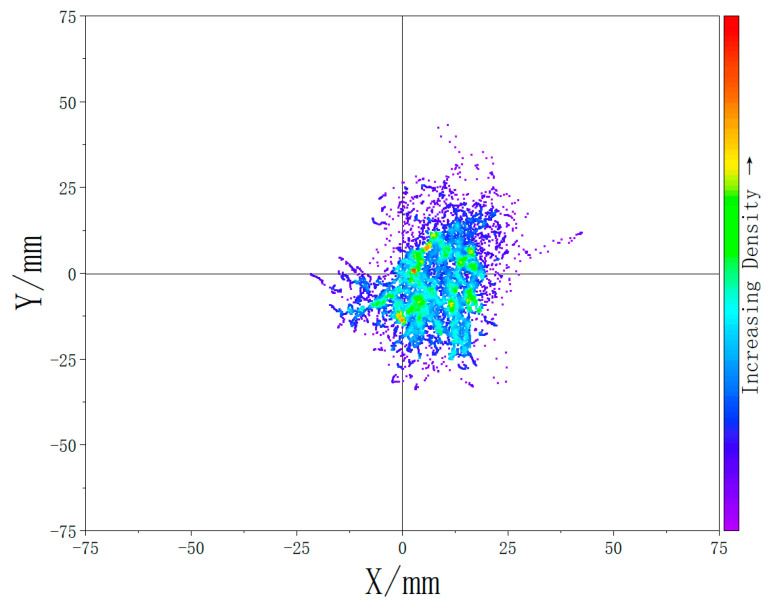
Plantar pressure center density map while wearing the exoskeleton chair (with extra support).

**Figure 19 biomimetics-10-00330-f019:**
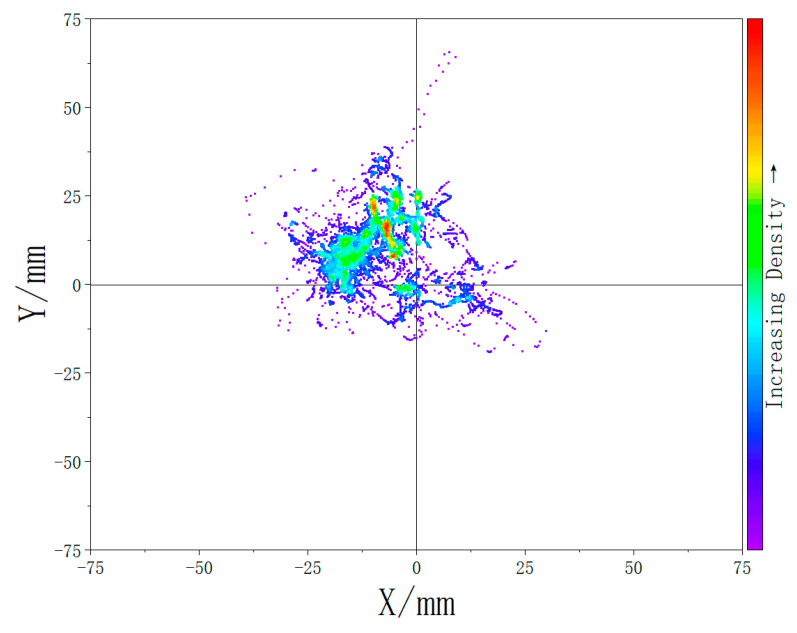
Plantar pressure center density map while wearing the exoskeleton chair (without extra support).

**Table 1 biomimetics-10-00330-t001:** Anthropometric data of Chinese adults aged 18–70 (partial) (in mm).

	Height	Body Weight	Thigh Length	Shank Length	Popliteal Height (Sitting)	Foot Length
Males	1773	83	506	405	442	264
Females	1650	70	476	375	408	243

**Table 2 biomimetics-10-00330-t002:** Comparative analysis of passive exoskeleton chair parameters.

Device	Development Stage	Side of Attaching	Main Material	Device Weight	Max Load	Muscle Load Reduction (EMG)
Medial-Support Exoskeleton Chair (Proposed Device)	Prototype	Medial	Al alloy	3.9 kg	100 kg	49.2~72.9%
Exoskeleton Chair 1 [27]	Commercialized	Lateral	-	6.2 kg	100 kg	57%
Exoskeleton Chair 2 [28]	Commercialized	Posterior	Polymers	3.3 kg	120 kg	64%
Exoskeleton Chair 3 [7]	Commercialized	Posterior	-	1.6 kg	80 kg (Small Size)	30.59~84.08%
Exoskeleton Chair 4 [29]	Prototype	Lateral	Metal	8.3 kg	130 kg	83%
Exoskeleton Chair 5 [30]	Prototype	Posterior	Al Alloy	1.5 kg	87 kg	49~77%

## Data Availability

The generated and analyzed datasets in the current study are not publicly available due to privacy concerns; however, the anonymized datasets are available from the corresponding author on reasonable request.

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
