# Peer review of "Research and Design of a Medial-Support Exoskeleton Chair"

_biomimetics, 2025, doi:10.3390/biomimetics10050330_

Round 1

Reviewer 1 Report

Comments and Suggestions for Authors

Pros
o The study addresses a very current and relevant problem: alleviating lower limb fatigue in workers who stand for long periods of time.
o The medial-support structure represents a significant departure from conventional lateral or posterior exoskeletons. This reduces the risk of collision and improves stability in confined workspaces.
o The authors use a combination of human motion capture, mathematical modeling, finite element analysis (FEA), and sEMG-based experimental validation, ensuring that their findings are supported by multiple methods.
o The reduction in muscle load, which ranges from 63.7% to 88.8% in different muscle groups, provides conclusive evidence for the effectiveness of the proposed system.
o The paper provides comprehensive mechanical and biomechanical models, which clearly describe the design logic and sizing of the exoskeleton parts.
o A functional prototype was built and tested, demonstrating practical feasibility.

Cons
o The experimental validation involved a very small number of participants, which limits the generalizability of the results.
o Although existing exoskeletons are briefly mentioned, there is limited quantitative comparison between the proposed design and commercially available or previously researched systems.
o The study focuses on short-term testing, with no information on long-term comfort, adaptability, durability, or user acceptability.
o Although informative, some diagrams (e.g., joint angle curves or CAD renderings) could benefit from clearer labeling or more explicit legends for better understanding.

Recommendations:
• It is recommended to conduct a larger user study involving more participants with different anthropometric characteristics (age, gender, height, weight, etc.). This will increase the generalizability and statistical robustness of the results.
• It is recommended to include a benchmark or performance comparison to existing lateral/posterior support exoskeletons, by including parameters such as:
o Weight
o Articulated torque support
o Muscle load reduction
o Comfort
o Cost
• It is recommended to include a section on ergonomics and user experience, evaluating:
o Comfort for prolonged use
o Ease of donning and doffing
o Maintenance requirements
o Potential skin pressure or circulation issues

Moreover, the authors should specify design constraints, testing limitations, and unresolved engineering challenges.

Reviewer 2 Report

Comments and Suggestions for Authors

The article titled "Design and Research of a medial support Exoskeleton Chair." may be retitled as : Research and Design of a medial support Exoskeleton Chair" to sound better. 

The article was aimed to address lower limb fatigue in a prolong standing workers and to develop a mechanism to reduce the fatigue in prolong standing worker. The article proposes a passive structural design for a medial supported exoskeleton seat. To accomplish the aim, key structural parameter of the medial support exoskeleton were investigated using torso lower limb system as a rigid system. Center of gravity projection, and support surface were analyzed. A sagittal-plane sit-to-stand model was employed on Ankle joint, Knee joint and total body mass to find optimal support angle for the wearer of exoskeleton, and support rod length to ensure length adjustment system, It was also verified using ANSYS (FEA) computer model. Subsequently a physical prototype of medial support exoskeleton chair was fabricated using 3D printing technology. It was later tested using eEMG testing using EMG sensors. Finally it was shown that when wearing the exoskeleton chair without support rod  muscle loads decreased by  44.6%, and when complete exoskeleton was wear with support rod, the muscle load further decreased by 65.0% and so on. However some field testing is required to support the claim. 

The article is logically and mathematically developed and analyzed with sound mathematically results presented. 

Conclusion needs further discussion to support the reduction in muscle load and physical experimentation/future guidelines to keep the research alive.  

Article needs to address some grammatical errors like avoid using first person, e.g., in line No. 72, We employed ....., may be better, using author employed....and so on.

Avoid using present tense e.g., in line No. 144, support surface is analzed, may be rewritten as "support surface was analysed...., and so on. 

Fig 2 and Fig. 3 captions may be make more clearer (Make proper spacing).

Reference No. 2 has no mention of year of publication.  

Comments on the Quality of English Language

Quality of English is satisfactory however needs some improvements with respect to grammatical errors. 

Round 2

Reviewer 1 Report

Comments and Suggestions for Authors

In my opinion, the paper can be published in present form.